# Environmental and genetic predictors of human cardiovascular ageing

Mit Shah[1], Marco H. de A. Inácio[1], Chang Lu[1], Pierre-Raphaël Schiratti[1], Sean L. Zheng[2], Adam Clement[1], Antonio de Marvao[1], Wenjia Bai[3,4], Andrew P. King[5], James S. Ware[1,2], Martin R. Wilkins[2], Johanna Mielke[6], Eren Elci[6], Ivan Kryukov[6], Kathryn A. McGurk[1,2], Christian Bender[6], Daniel F. Freitag[6] & Declan P. O'Regan[1] ✉

Cardiovascular ageing is a process that begins early in life and leads to a progressive change in structure and decline in function due to accumulated damage across diverse cell types, tissues and organs contributing to multi-morbidity. Damaging biophysical, metabolic and immunological factors exceed endogenous repair mechanisms resulting in a pro-fibrotic state, cellular senescence and end-organ damage, however the genetic architecture of cardiovascular ageing is not known. Here we use machine learning approaches to quantify cardiovascular age from image-derived traits of vascular function, cardiac motion and myocardial fibrosis, as well as conduction traits from electrocardiograms, in 39,559 participants of UK Biobank. Cardiovascular ageing is found to be significantly associated with common or rare variants in genes regulating sarcomere homeostasis, myocardial immunomodulation, and tissue responses to biophysical stress. Ageing is accelerated by cardio-metabolic risk factors and we also identify prescribed medications that are potential modifiers of ageing. Through large-scale modelling of ageing across multiple traits our results reveal insights into the mechanisms driving premature cardiovascular ageing and reveal potential molecular targets to attenuate age-related processes.

Cardiovascular disease (CVD) is the leading cause of death globally, and ageing is a primary risk factor for its development and progression[1,2]. Cardiovascular ageing is a process that begins in early life and occurs at multiple scales across different organ systems leading to an accumulation of damage that cannot be recovered through endogenous repair and regeneration. Biophysical, metabolic and immunological factors lead to a pro-fibrotic state, cellular senescence, and end-organ damage affecting both the heart and circulatory system[3,4]. Ageing is driven by intrinsic processes that act at genetic, molecular and cellular targets, as well as extrinsic drivers such as

lifestyle and environmental risk factors that modify these processes. In common with ageing in other organ systems, these mechanisms converge upon dysregulated inflammation, alteration of epigenetic modifications, and metabolic imbalances[5]. A final common pathway of cardiovascular ageing is loss of tissue compliance which is manifest through diastolic dysfunction, interstitial fibrosis and vascular remodelling[6,7]. Such changes can be assessed through non-invasive imaging and enable an estimate of how an individual's cardiovascular system has aged relative to a normative population[8]. An equivalent calculation of the 'age gap' has been shown to be a promising neuro-

[1]MRC London Institute of Medical Sciences, Imperial College London, London, UK. [2]National Heart and Lung Institute, Imperial College London, London, UK. [3]Department of Computing, Imperial College London, London, UK. [4]Department of Brain Sciences, Imperial College London, London, UK. [5]School of Biomedical Engineering & Imaging Sciences, King's College London, London, UK. [6]Bayer AG, Research & Development, Pharmaceuticals, Wuppertal, Germany. ✉e-mail: declan.oregan@imperial.ac.uk

imaging marker for modelling dynamic lifespan trajectories of brain ageing[9–11].

Here we used computer vision techniques to analyse cardiac magnetic resonance (CMR) imaging in 39,559 participants of the UK Biobank to extract image-derived phenotypes of the three-dimensional (3D) geometry and motion of the heart as well as dynamic assessment of vascular function. These image-derived traits were used to train supervised machine learning algorithms to predict participants' ages and derive a cardiovascular age-delta for each individual, quantifying the deviation in years from healthy ageing. We also trained a model to learn the spatial features of electrocardiograms (ECG) associated with aging. Using age-delta as a trait for genome-wide common and exome-wide rare variant association analyses, we found ageing-associated loci containing genes known to be associated with tissue elasticity, myocardial contractility, inflammatory regulation and immune response to apoptosis. We also describe the relationship between cardiovascular risk factors and characteristics of premature cardiovascular ageing, and describe associated patterns of structural, functional and tissue-level changes in the heart. Together these analyses reveal the environmental and genetic mechanisms which underlie ageing of the cardiovascular system and indicate potential targets for risk modification.

## Results

### Study overview

We analysed CMR data from 39,559 participants in UK Biobank using machine learning segmentation and motion tracking to measure multiple imaging traits associated with cardiovascular structure, function and fibrosis (Fig. 1). Baseline characteristics of the population and a flow chart of analysis steps are shown in Supplementary Materials. We first trained a machine learning model to define healthy cardiovascular ageing in a development set determined to be free of cardiac, respiratory and metabolic disease, using a gradient boosting algorithm (CatBoost)[12]. We used the image-derived traits to predict cardiovascular age and computed an age-delta for the difference between predicted age and chronological age. We then predicted cardiovascular age in the rest of the UK Biobank population, and analysed the associations of the cardiovascular age-delta with traditional cardiovascular risk factors. We next performed a genome-wide association study (GWAS) of cardiovascular age-delta and then a phenome-wide association study (PheWAS) to identify phenotypes associated with both age-delta and polygenic risk score (PRS) for age-delta. We performed rare variant association analyses using whole exome sequencing (WES). Latent features of ECG traits may also be associated with ageing, and so we also trained a deep learning network to predict corresponding "ECG-ages" and age-deltas to discover any shared genetic architecture with cardiovascular ageing[13].

### Image and electrocardiogram phenotyping

We performed automated quality-controlled analysis on CMR cine imaging to assess bi-atrial and bi-ventricular volumes and function, as well as left ventricular mass[14]. Diastolic function, which is a key feature of the ageing heart, was assessed using motion analysis to derive end-diastolic strain rates[6]. We assessed diffuse myocardial fibrosis, an early feature of natural ageing[15], using native T1 mapping of the inter-ventricular septum[16]. Central vascular function was assessed by measuring aortic distensibility from central blood pressure estimates and dynamic aortic imaging[14]. In total, 126 quantitative imaging phenotypes characterising structure, function and tissue characteristics were generated for each participant. To visualise variation in cardiac morphology with age, we used time-resolved 3D morphometry of the heart[17]. Resting ECG traces were parsed from XML files in UK Biobank. Each ECG sample corresponded to a measured recording for 1 s at 500 Hz (see Supplementary Materials for further pre-processing details).

### Cardiovascular and ECG age predictions

A CatBoost machine learning model trained on healthy participants ($n = 4019$), applied on a holdout test set ($n = 1044$), yielded a coefficient of determination ($R^2$) of 0.49, a Pearson correlation coefficient ($|r|$) between predicted age and chronological age of 0.70 ($P < 1.0 \times 10^{-16}$) and a mean absolute error (MAE) of 4.21 years. After bias-correction, there was no correlation between cardiovascular age-delta and chronological age ($|r| = -5.5 \times 10^{-16}$, $P \approx 1$), showing that any deviations from healthy cardiovascular ageing were not related to the participant's actual age. There was a strong correlation between predicted age and chronological age ($|r| = 0.85$, $P < 2.2 \times 10^{-16}$) using latent ECG features, which is comparable to other deep learning architectures[18]. There was also no relationship to the participants' actual age ($|r| = -0.01$, $P \approx 1$).

The distribution and correlation between image-derived traits is shown in Figs. 2, 3, and the feature importance of traits for cardiovascular age-delta is shown in Fig. 4. Differences between sexes were observed across most phenotypes and were strongest for volumetric data. Sex was therefore used as a covariate in all analyses. There were also correlations between left and right chamber measurements,

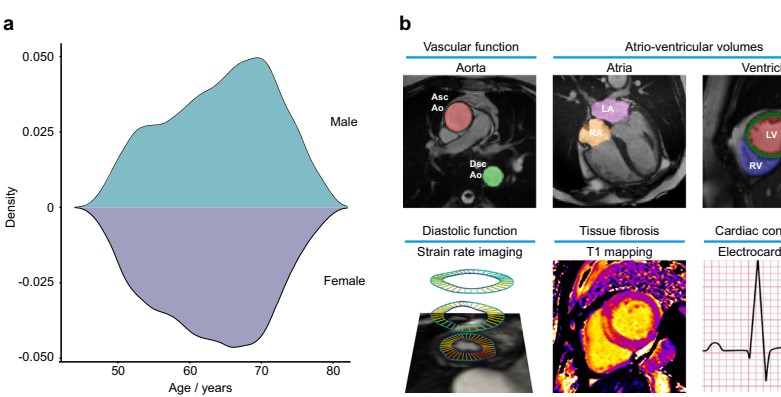
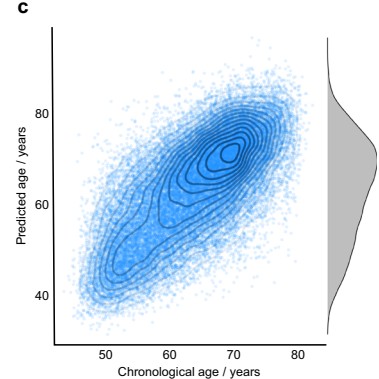

**Fig. 1 | Summary of data used for cardiovascular age prediction in UK Biobank. a** Age distributions of participants by sex (kernel density estimates, 20,502 females, and 18,947 males). **b** Phenotypes used for prediction. The top row shows cardiac magnetic resonance images with automated time-resolved segmentation of the aorta and cardiac chambers. The bottom row shows an example of left ventricular motion analysis to derive radial strain rate and a parametric T1 map of the left ventricular myocardium. Resting electrocardiograms (ECGs) were used in an independent model for age prediction. LA left atrium, LV left ventricle, RA right atrium, RV right ventricle, Asc Ao ascending aorta, Dsc Ao descending aorta. **c** The relationship between predicted and chronological cardiovascular age ($n = 34,137$, ages jittered, density contours, and a marginal density plot.).

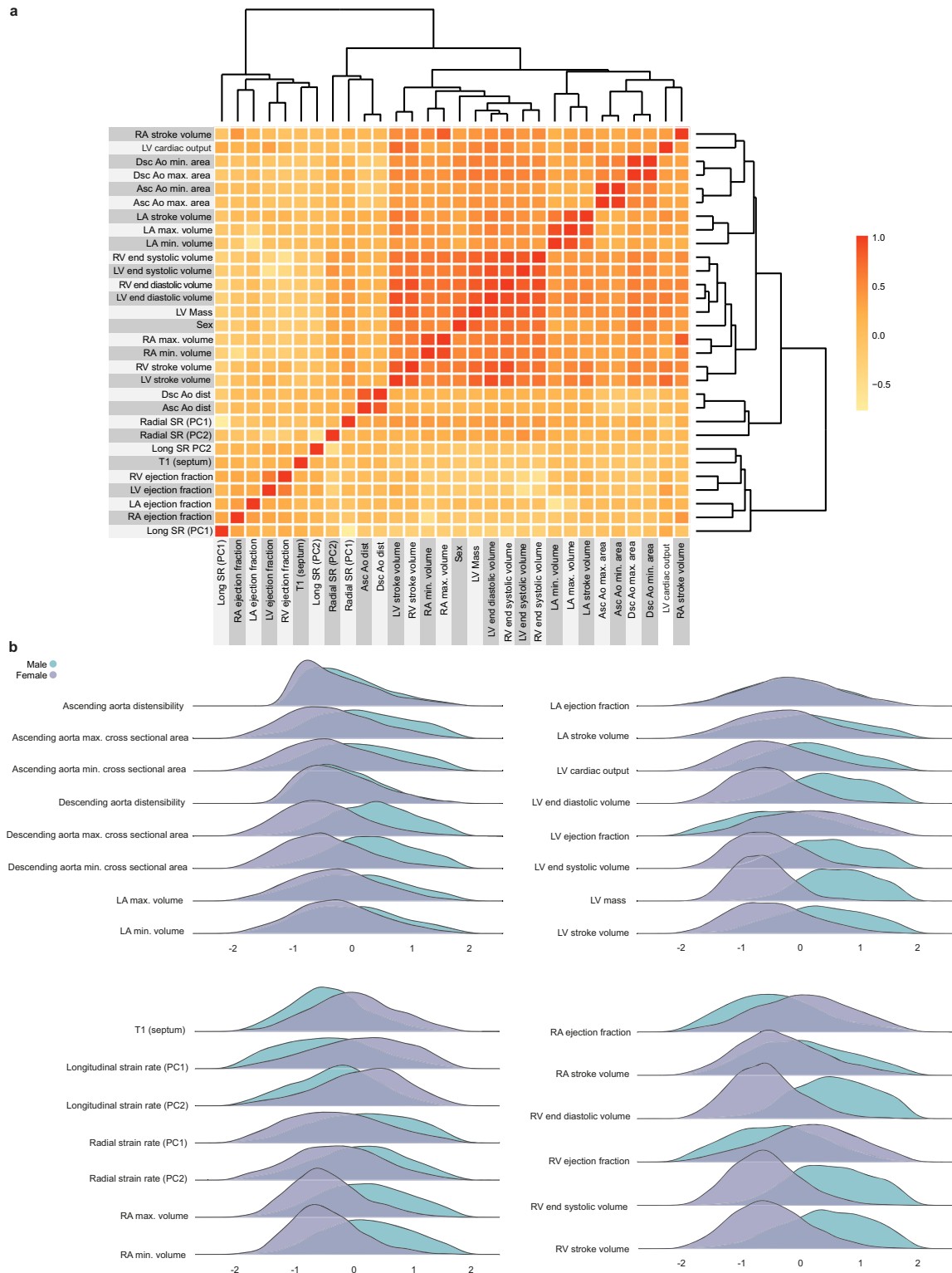

**Fig. 2 | A summary of non-imaging and imaging features used in the cardiovascular age prediction model. a** Heatmap for the features, with colours representing the Pearson correlation coefficient ($n = 39{,}559$). **b** Ridge plots summarising the distribution densities of features, normalised for visualisation purposes ($n = 39{,}559$). Asc Ao dist ascending aortic distensibility, Asc Ao min./max. area ascending aortic minimum/maximum cross-sectional area, Dsc Ao dist descending aortic distensibility, Dsc Ao min./max. area descending aortic minimum/maximum cross-sectional area, LA left atrium LASV left atrial stroke volume, LAEF left atrial ejection fraction, LVESV left ventricular end-systolic volume, LVEDV left ventricular end-diastolic volume, LVCO left ventricular cardiac output, LVM left ventricular mass, LV left ventricle, PC principal component, RA right atrium, RV right ventricle, Radial/Long SR radial/longitudinal strain rates (numbers in bracket referring to frame number in cardiac cycle), RA max. vol right atrial maximum volume, RVESV right ventricular end-systolic volume.

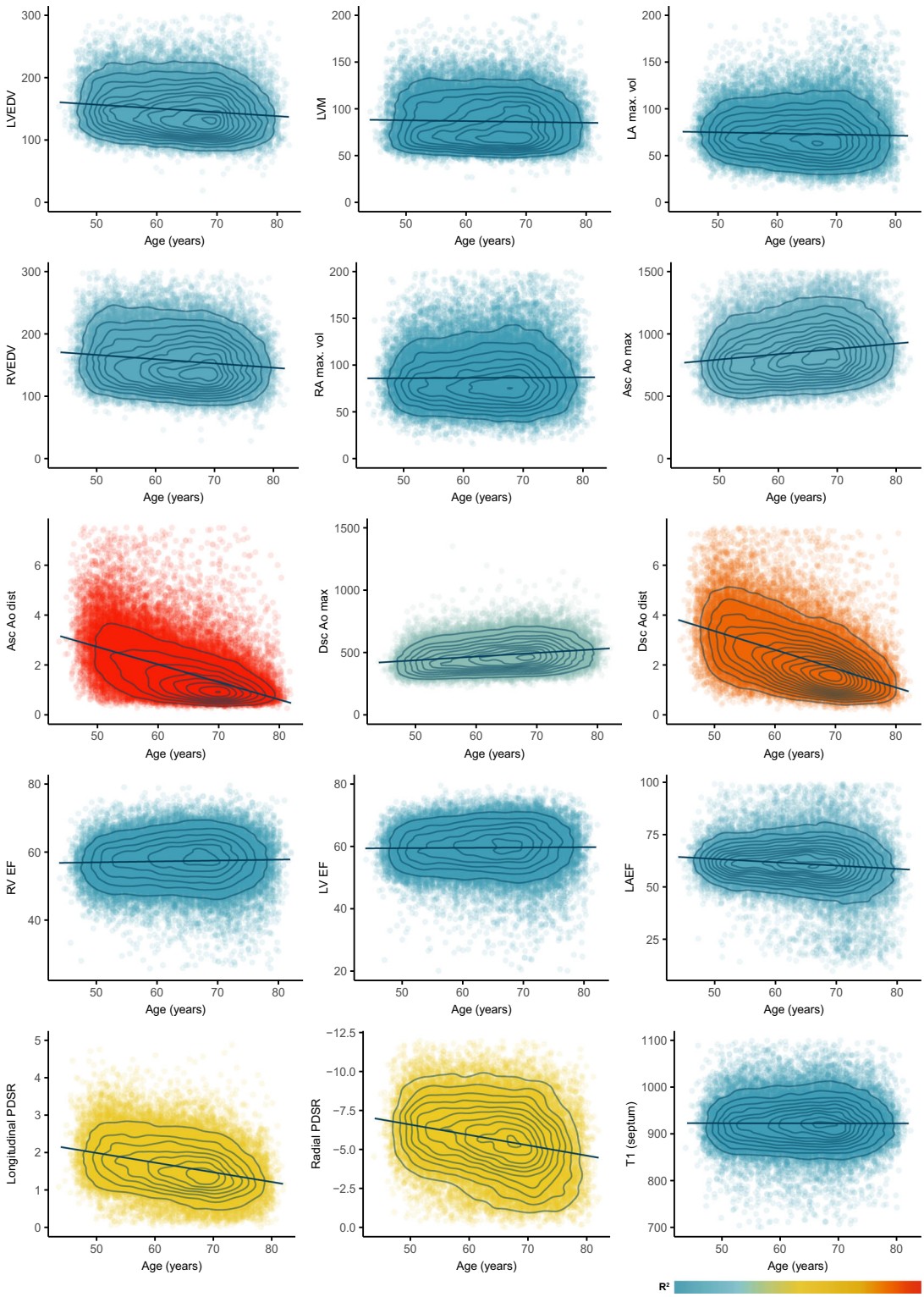

**Fig. 3 | Image phenotype associations with chronological age.** In total, we measured 126 image-derived phenotypes, including temporal motion analysis of cine imaging. A selection of 15 representative phenotypes of volumes, function, and tissue characterisation are shown with their relationship to chronological age at the time of imaging. ($n = 39,443$, ages jittered, density contours, point colours represent coefficient of determination ($R^2$)). LVEDV left ventricular end-diastolic volume, LVM left ventricular mass, LA max. vol left atrial maximum volume, RVEDV right ventricular end-diastolic volume, RA max. vol right atrial maximum volume, Asc/ Dsc Ao max. ascending/descending aortic maximal cross-sectional area, Asc/Dsc Ao dist. ascending/descending aortic distensibility, RV/LVEF right/left ventricular ejection fraction, LAEF left atrial ejection fraction, PDSR peak diastolic strain rate, T1 longitudinal relaxation time of the tissue.

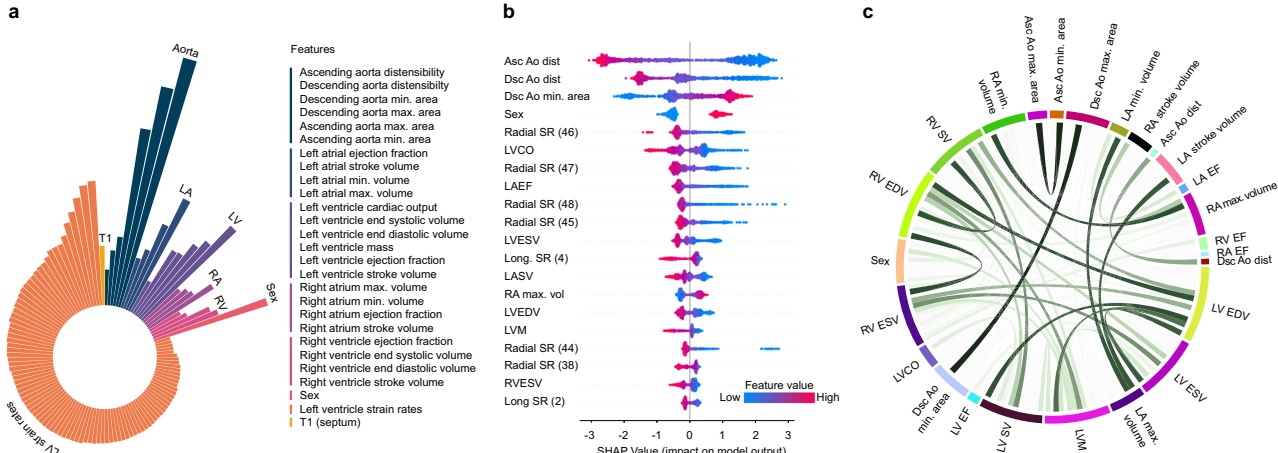

**Fig. 4 | Features associated with cardiovascular ageing. a** Radial plot of log-transformed, normalised feature importance in cardiovascular age prediction using CatBoost grouped by category (aortic structure and function (Aorta), left atrial (LA) and left ventricular (LV) structure and function, right atrial (RA) and right ventricular (RV) structure and function, sex, left ventricular strain rates and myocardial native septal T1. **b** SHAP (Shapley additive explanations) plot of the top twenty features contributing to cardiovascular age prediction. The colour represents the feature value (red high, blue low), and its contribution to model prediction output. Features include ascending (Asc Ao) and descending aortic (Dsc Ao) distensibility (dist), descending aortic minimum cross-sectional area (Dsc Ao min. area), sex, radial and longitudinal strain rates (Radial SR, Long SR, numbers in bracket referring to frame number in cardiac cycle), left atrial stroke volume (LASV), left atrial ejection fraction (LAEF), left ventricular end-systolic and diastolic volume (LVESV, LVEDV), left ventricular cardiac output (LVCO), left ventricular mass (LVM), right atrial maximum volume (RA max. vol) and right ventricular end-systolic volume (RVESV). **c** Circos plot of the correlation between imaging features. Ribbon widths are proportional to the absolute value of the Pearson correlation coefficient (|*r*|). For simplicity and clarity, absolute correlations are hidden where |*r*| < 0.4, and between radial and longitudinal strain measures. EDV end-diastolic volume, ESV end-systolic volume, EF ejection fraction, SV stroke volume. All plots n = 34,137.

reflecting ventricular interdependence, with left-sided traits having greater predictive value. Arterial distensibility, in both the ascending and descending thoracic aorta, was the strongest predictor of age-delta. Late diastolic strain rates were also important contributors to the predictive model, especially the radial component. 3D models of the heart also enabled visualisation of the morphological and functional changes related to age (Fig. 5). Left ventricular volumes decreased with age with the remodelling of the lateral wall, while wall thickness showed progressive septal hypertrophy. Motion analysis demonstrated that regional changes in both left ventricular systolic contraction and diastolic relaxation occur with ageing.

### Phenome-wide association study and risk factor analysis
We first aimed to assess associations of cardiovascular age-delta with an unbiased range of diseases in the UK Biobank using a PheWAS. Here we performed independent logistic regression analyses with age-delta as a predictor for each outcome adjusted for age, sex, ethnicity and multiple comparisons. We used PheCodes to aggregate a broad range of disease classifications and showed that age-delta was predominantly associated with circulatory and metabolic disorders (Fig. 6a), with the strongest associations linked to hypertension and diabetes.

We next aimed to quantify the associations between established risk factors and cardiovascular age-delta. We used linear regression with each risk factor in turn as a predictor and age-delta as the dependent variable. We adjusted for age, sex and multiple comparisons controlling for false discovery and showed that hypertension (+1.58 years, $P < 1.0 \times 10^{-16}$) and diabetes (+0.74 years, $P = 0.0012$) were both associated with an increase in predicted age (Fig. 6b).

Age-delta was also elevated in both males with obesity (+0.46 years, $P = 0.018$), and females with coronary artery disease (+0.85 years, $P = 0.047$). Serum levels of apolipoprotein B (+0.70 years per mg/dL, $P = 5.0 \times 10^{-4}$), triglyceride (+0.56 years per mmol/L, $P < 1.0 \times 10^{-16}$) and LDL (+0.13 years per mg/dL, $P = 0.020$) levels were all also associated with adverse effects on cardiovascular ageing.

Smoking (+0.03 years per smoking pack year, $P = 3.0 \times 10^{-9}$) and alcohol (+0.02 years per gram per day increase in daily alcohol consumption, $P = 2.5 \times 10^{-8}$) were also associated with adverse effects on

cardiovascular ageing, with comparable effect sizes to brain ageing[19]. Telomere length was associated with favourable effects on cardiovascular age (−0.10 years per unit increase in z-adjusted telomere length, $P = 0.025$) (Fig. 6c).

### Age-delta and cardiovascular events
We found a modest association between cardiovascular age-delta and major adverse cardiovascular events (hazard ratio, 1.09; 95% confidence intervals, 1.01−1.21; $P = 0.022$) when comparing upper and lower quartiles of age-delta in a covariate-adjusted model, but no significant associations in models considering cardiovascular age-delta as a continuous variable (see Supplementary Material for details).

### Prescribed medications and cardiovascular age
To assess the potential use of age-delta as a surrogate marker of cardiovascular disease progression, we also performed separate drug-specific linear regression models with age-delta as the dependent variable, and drug use as the independent variable. We used two models, with the first model adjusting for sex, age, $age^2$, cardiovascular risk factors and the second model additionally adjusting for haemodynamics (blood pressure and pulse rate) (see Supplementary Material for further details). Most associations between anti-hypertensives and age-delta were explained by cardiovascular risk factors and haemodynamics. Beta-blocker (+0.61 years) and metformin (+0.80 years) use was associated with elevated cardiovascular age, and calcium channel blocker (−0.34 years) use was associated with favourable cardiovascular age, independent of cardiovascular risk factors and haemodynamic parameters (Fig. 7).

### Genome-wide association study of cardiovascular age-delta
All genetic analyses are reported in compliance with STREGA guidelines[20]. The proportion of phenotypic variance in image-derived age-delta due to additive genetic variation ($h^2$) explained by all genotyped single nucleotide polymorphisms (SNPs) was 10.5% (see Supplementary Material). We identified five genome-wide significant independent loci from our GWAS that are associated with image-derived cardiovascular age-delta ($P = 5 \times 10^{-8}$) (Fig. 8a). Summary

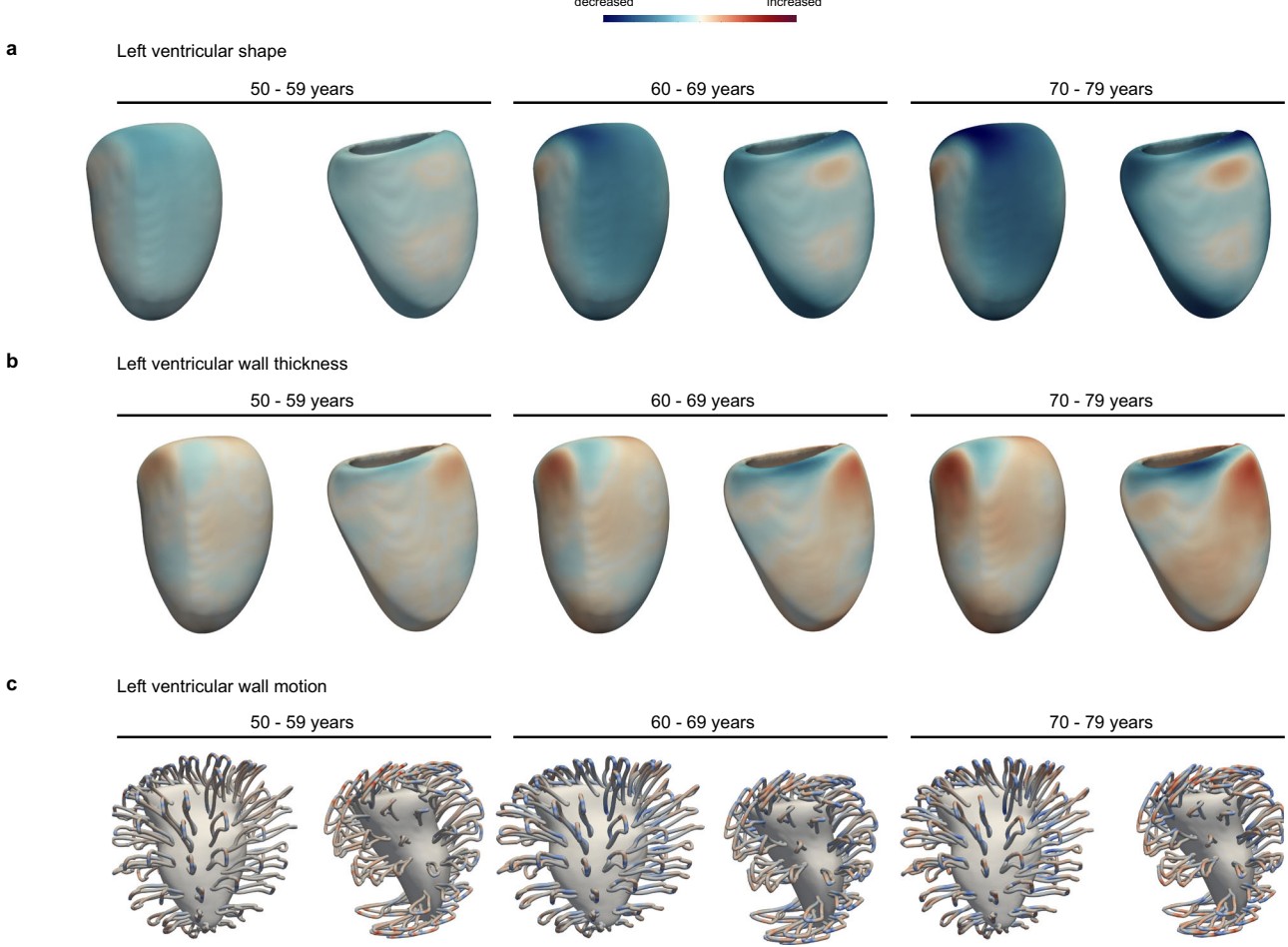

**Fig. 5 | Three-dimensional models of cardiac ageing.** Three-dimensional mapping of left ventricular shape (**a**), left ventricular wall thickness (**b**) and left ventricular motion (**c**) with increasing age. The models show the mean phenotype for each decade of age relative to 40–49-year olds, aggregating data using registration of cardiac segmentations, with parameters represented on the epicardial surface. Paired views of the inferolateral and anteroseptal walls.

information for the 5 loci identified using the full GWAS dataset are presented in Table 1, with further information provided in Supplementary Material. The nearest gene to the locus is defined, along with the most "likely gene" based on: evidence of a functional effect on a gene; previously documented cardiovascular disease association; or reported mechanism potentially involved in cellular ageing processes (Supplementary Material). The lead variants have known roles in myocardial contractility (Titin, *TTN*)[21], and arterial mechanics (Elastin, *ELN*)[22], and have been implicated in pro-inflammatory activity and antihypertensive responses (Phospholipase C Epsilon 1, *PLCE1*)[23,24]. Additionally, *TTN* was a likely causal gene for ECG-derived age-delta amongst 7 other independently-associated loci that include genes related to trabecular development (T-Box Transcription Factor 3, *TBX3*)[25], and regulation of cardiac rhythm (Sodium voltage-gated channel alpha subunit 5, *SCN5A*; Calcium/Calmodulin Dependent Protein Kinase II Delta, *CAMK2D* and Myozenin 1, *MYOZ1*) (Fig. 8b).

A covariate-corrected polygenic risk score (PRS) comprising 13 SNPs selected using a clumping/thresholding approach for image-derived cardiovascular age-delta was evaluated in 373,948 independent genotyped participants of UK Biobank, and showed an association with hypertension ($P = 2.9 \times 10^{-12}$) (see Supplementary Material).

### Rare variant association study of cardiovascular age-delta

Predicted loss of function variant (pLoF) gene burden testing using Regenie[26], with three variant prioritisation masks on allele frequencies applied, identified two further genes associated with image-derived cardiovascular age-delta independent of common variant signals. These were Triggering Receptor Expressed On Myeloid Cells 2 (*TREM2*) and Mitochondrial Calcium Uptake Family Member 3 (*MICU3*). *TREM2* encodes a transmembrane receptor expressed in the central nervous system and macrophages, and has been linked to anti-inflammatory effects in atherosclerotic plaque, cardiac tissue repair and immunomodulation of monocytes in the heart[27–29]. *MICU3* encodes a uniporter channel in the heart and skeletal muscle, which regulates calcium homoeostasis[30,31].

## Discussion

Cardiovascular ageing is an interaction of genetic, cellular, and biophysical processes that results in declining adaptive homoeostasis. In this study, we used machine learning of multiple image-derived and electrical cardiovascular phenotypes to predict biological age and determine the genetic and environmental associations with deviation from healthy ageing. We found that healthy ageing was associated with a progressive decline in both vascular and myocardial tissue compliance, and these traits were also the strongest predictors of deviation from healthy cardiovascular ageing. A decline in diastolic function is recognised as a hallmark of cardiac ageing, which occurs through multiple pro-fibrotic and energetic pathways.[32,33] A key driver of diastolic impairment is myocardial interstitial fibrosis, and we found that an imaging biomarker of fibrosis predicted accelerated ageing. Variation in this specific biomarker is thought to be mediated by glucose metabolism, tissue repair and oxidative stress[34], with fibrosis

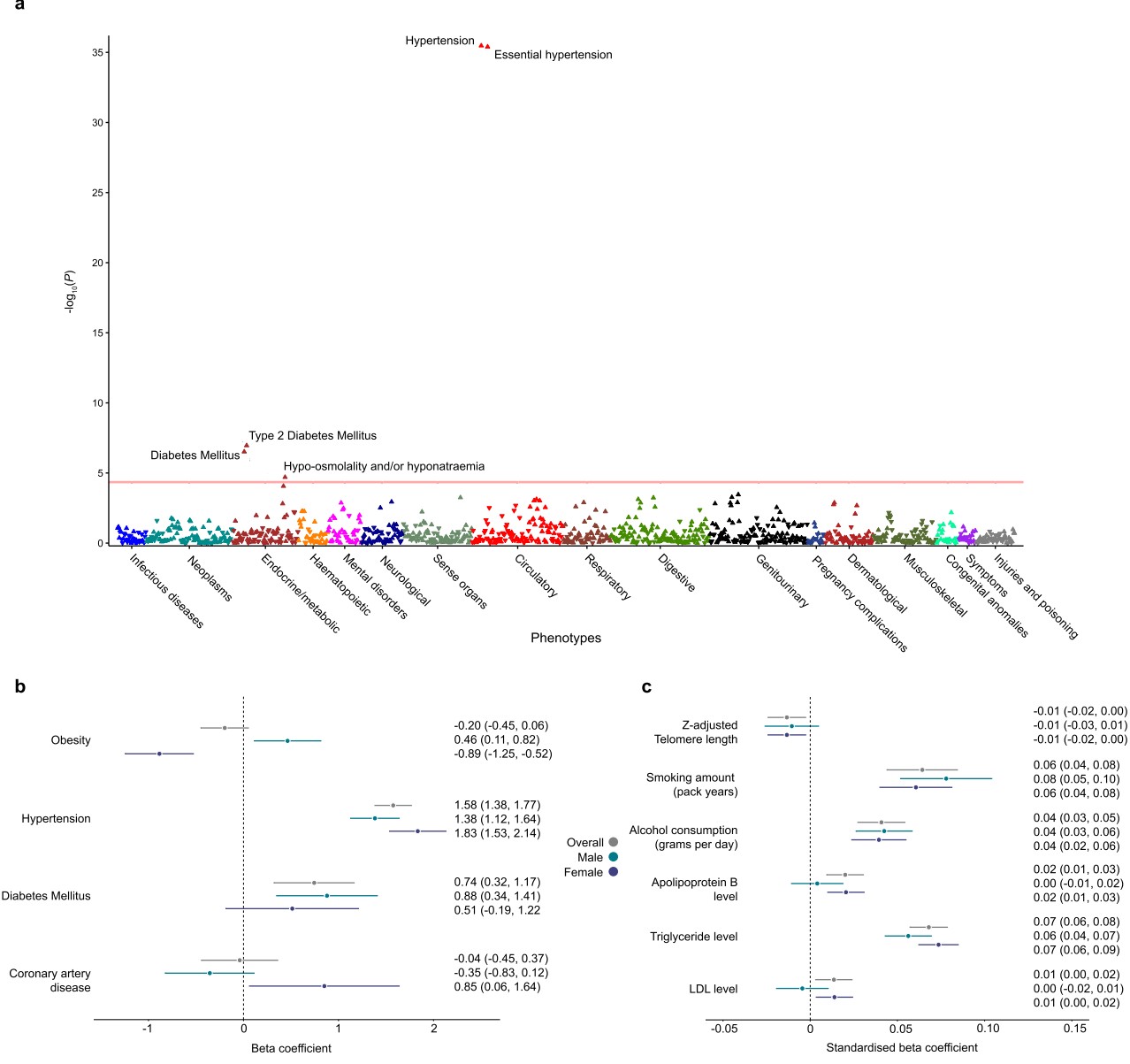

**Fig. 6 | Cardiovascular age-delta PheWAS and risk factor associations.**
**a** Phenome-wide analysis of cardiovascular age-delta adjusted for age, age$^2$, sex and the first ten genetic principal components by two-sided logistic regression ($n = 34{,}137$ participants). The red line represents the significance threshold after accounting for multiple testing using Bonferroni correction (1149 phenotypes, $P < 4.4 \times 10^{-5}$). Upright triangles indicate positive correlations, and inverted triangles indicate negative correlations. **b** Linear regression analysis of categorical risk factors with cardiovascular age-delta. Error bars indicate the beta-coefficients point estimates ± 95% confidence intervals (CI) of the experiments, adjusted for age, age$^2$ and sex, and compared with age and sex-matched controls. The test samples comprised $n = 7089$ participants with obesity, $n = 7089$ controls without obesity;

$n = 11{,}047$ participants with hypertension, $n = 11{,}047$ controls without hypertension; $n = 2466$ participants with diabetes mellitus (DM), $n = 2466$ controls without DM; $n = 2658$ participants with coronary artery disease (CAD), $n = 2658$ controls without CAD. **c** Linear regression analysis of quantitative risk factors with cardiovascular age-delta. Error bars indicate the beta-coefficients point estimates per standard deviation (SD) increase in a unit of risk factor ± 95% confidence intervals (CI) of the experiments, adjusted for age, age$^2$ and sex. Data comprises 32,151 participants with apolipoprotein B data, 32,263 participants with triglyceride data and 32,241 participants with low-density lipoprotein (LDL) data, 31,871 participants with telomere length data, 9283 participants with smoking data and 21,374 participants with alcohol consumption data.

being a final common pathway after homoeostatic mechanisms are exhausted[35]. Three-dimensional image analysis showed that ageing results in asymmetrical remodelling, and patterns of contractility and relaxation in the left ventricle. We observed progressive septal hypertrophy, which is recognised as an independent feature of ageing in longitudinal studies[36]. The aorta also becomes stiffer with age, which is associated with numerous cardiovascular disease endpoints[37]. Elastin fragmentation, endothelial dysfunction and deposition of advanced glycation end-products are thought to play a causal role[38], and distensibility is associated with pathways related to cardiovascular

development, extracellular matrix production, and smooth muscle cell contraction[39].

Our study provides insights into the biological basis of heterogeneity in cardiovascular ageing. One of our loci implicated *PLCE1*, which regulates contractile myocardial reserve with loss of expressed PLCε signalling sensitising the heart to the development of hypertrophy in response to chronic cardiac stress[40]. PLCε also regulates inflammatory responses to myocardial injury[23], consistent with the putative causal relationship of 'inflammageing' with cardiovascular disease and multi-morbidity[41]. *ELN* was also a leading locus, with elastin

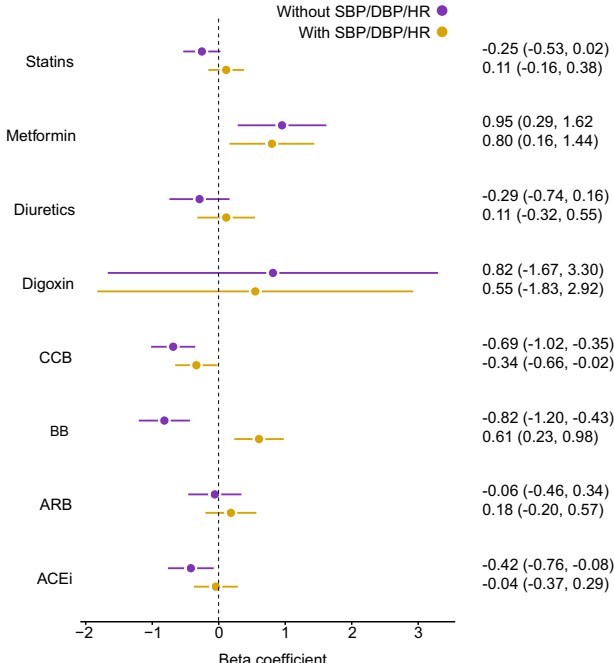

**Fig. 7 | Cardiovascular age-delta and drug use associations.** Linear regression analysis of cardiovascular drug usage with cardiovascular age-delta. Error bars indicate the beta-coefficients point estimates ± 95% confidence intervals (CI) of the experiments ($n$ = 27,546 participants). The results of the two models are demonstrated. One model adjusted for age, age² and sex, cigarette and alcohol intake, body-mass index, diagnoses of obesity, coronary artery disease, hypertension, diabetes mellitus, hypercholesterolaemia and heart failure. A second model additionally adjusted for haemodynamic parameters (heart rate, diastolic blood pressure and systolic blood pressure). ACEi angiotensin-converting enzyme inhibitors, ARB angiotensin receptor blockers, BB beta-blockers, CCB calcium channel blockers, DBP diastolic blood pressure, HR heart rate, SBP systolic blood pressure.

being an abundant extracellular matrix protein that provides elasticity and resilience to tissues, including the distensibility of blood vessels. The low turnover of elastin and chronic biophysical stress makes it susceptible to age-related changes caused by mechanical fracture, proteolysis, and calcification[22]. Age-associated arterial structural and functional decline may not be inevitable and lifestyle interventions have been shown to preserve vascular elasticity[42]. We observed two loci in common between our cardiovascular age-delta GWAS and previously reported single-trait GWASs of aortic function, suggesting that the majority of these trait-associated pathways are not modifiers of ageing[39,43]. For instance, elastin expression and organisation modify arterial ageing through signalling pathways which are accelerated by oxidative stress and inflammation.[41,44] Using WES, we found rare variants in two further genes that were associated with deviation from healthy cardiovascular ageing. The protein expressed by *TREM2* is an immune regulator controlling monocyte/macrophage transitions in response to apoptosis in the heart[28]. This suggests it may play a role at the interface of immunomodulation and cardioprotection in the ageing heart. *MICU3* plays an important role in regulating pathological calcium overload in the heart[30], and has been implicated in determining antioxidant responses in skeletal muscle ageing[31].

We identified variants in a cardiomyopathy-associated gene (*TTN*) suggesting a role for sarcomere homoeostasis as a mediator of human cardiac ageing. This may occur through an age-dependent effect on adaptive capacity that is distinct to the changes in titin isoform expression seen in heart failure[45]. Common variants in *TTN* were also implicated in ECG age-delta, suggesting a shared genetic mechanism with conduction-associated traits, however, other loci were predominately related to pathways regulating cardiac rhythm, suggesting

that the conduction system ages through distinct mechanisms, as has been observed in other organs and tissues[46]. For instance, *CAMK2D* is a gene associated with atrial remodelling, fibrosis and premature atrial fibrillation, and we found it to be a factor influencing ageing in the conduction system[47]. Fibrosis due to ageing is thought to play a major role in modulating conduction and repolarization diseases, with age also known to affect the severity and expressivity of the channelopathy-associated gene *SCN5A* identified in the ECG age-delta GWAS[48]. Gain or loss of function mutations in *SCN5A* are also implicated in progressive atrial fibrosis, although the mechanisms are not fully understood[49].

Overall the SNP-based heritability of cardiovascular ageing was 10.5%, suggesting that non-genetic contributions predominantly affect ageing. We found that cardiometabolic risk factors, including hypertension, diabetes and dyslipidemia, all contribute to an increased age-delta. 'Early vascular ageing' is thought to be induced by the integrated effect of haemodynamic factors, glycaemic dysregulation and fetal programming[50], while diabetes and dyslipidaemia have also been shown to be causally associated with diastolic dysfunction, which is also a major contributor to premature ageing in the heart[6]. Atherosclerosis is associated with normal and premature biological ageing, and although we could not assess atherogenesis or plaque burden directly, our image-derived phenotypes are sensitive to downstream effects of arteriosclerosis and vasculopathy as plaque is associated with diffuse changes to distensibility[51]. The lack of specific atherosclerotic phenotyping may underestimate its role in mediating ageing, but we did find that circulating atherosclerosis-related biomarkers were associated with accelerated ageing.

Obesity showed divergent effects in each sex, and in men, was associated with accelerated ageing while the opposite effect was observed in women. Obesity and overweight are associated with a non-linear increased risk of all-cause mortality[52], but misclassification of body composition, particularly among women, may lead to biased risk estimates[53]. There is evidence that obesity accelerates the ageing process through epigenetic alterations, mitochondrial dysfunction, cellular senescence and a pro-inflammatory state[54]. Obesity may also affect telomere dynamics and accelerate the ageing process, although the data is heterogeneous[55]. We did observe an association between telomere shortening and premature cardiovascular ageing, although the effect size was relatively small. Alcohol and smoking showed adverse associations with cardiovascular ageing that were of similar effects sizes to those observed in brain ageing[19]. For cardiovascular age-delta to be a useful surrogate endpoint, it would need to be correlated with hard clinical outcomes in the absence of intervention and modulated by established therapies, with its magnitude of modulation related to the effect of the intervention[56]. In cross-sectional analyses, whilst most antihypertensive effects on ageing were explained by known risk factors, beta-blockers, CCB, and metformin were independent predictors of age-delta. Further research is needed to investigate the response of cardiovascular age-delta to drug therapy, either in appropriately designed interventional studies, or through analysis of emerging observational data, whilst appropriately accounting for potential biases in this type of data.

There are limitations of this study. The rate of participation in the UK Biobank is higher among women, older age groups, and persons living in less socioeconomically deprived areas[57]. The population is predominantly European and further work is required to explore ageing traits and outcomes in people of diverse ancestries and social groups as an accelerated ageing phenotype may be observed due to the interaction of biological, psychosocial and socioeconomic factors[58]. Cardiovascular age-delta is derived at a single time-point in this cross-sectional study, and we could not assess within-person ageing of the cardiovascular system nor fully account for differential cohort and periodic effects. Nonetheless, such studies are able to capture disease-relevant features of biological ageing that are not

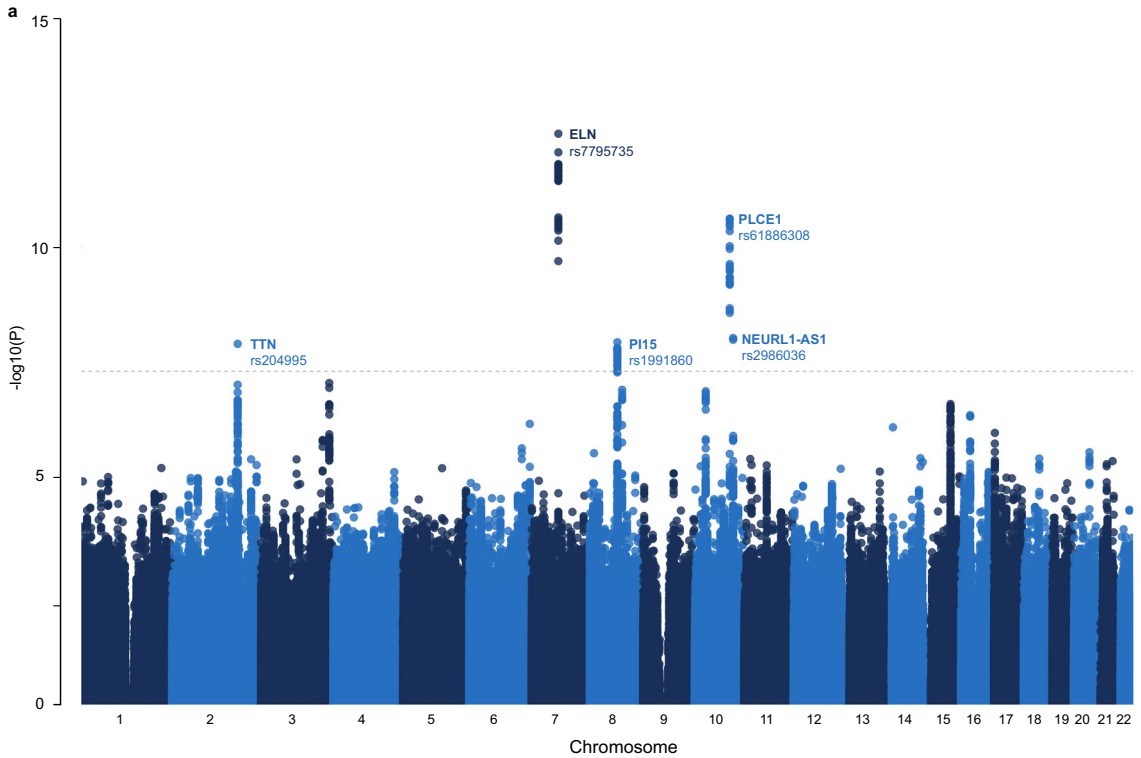

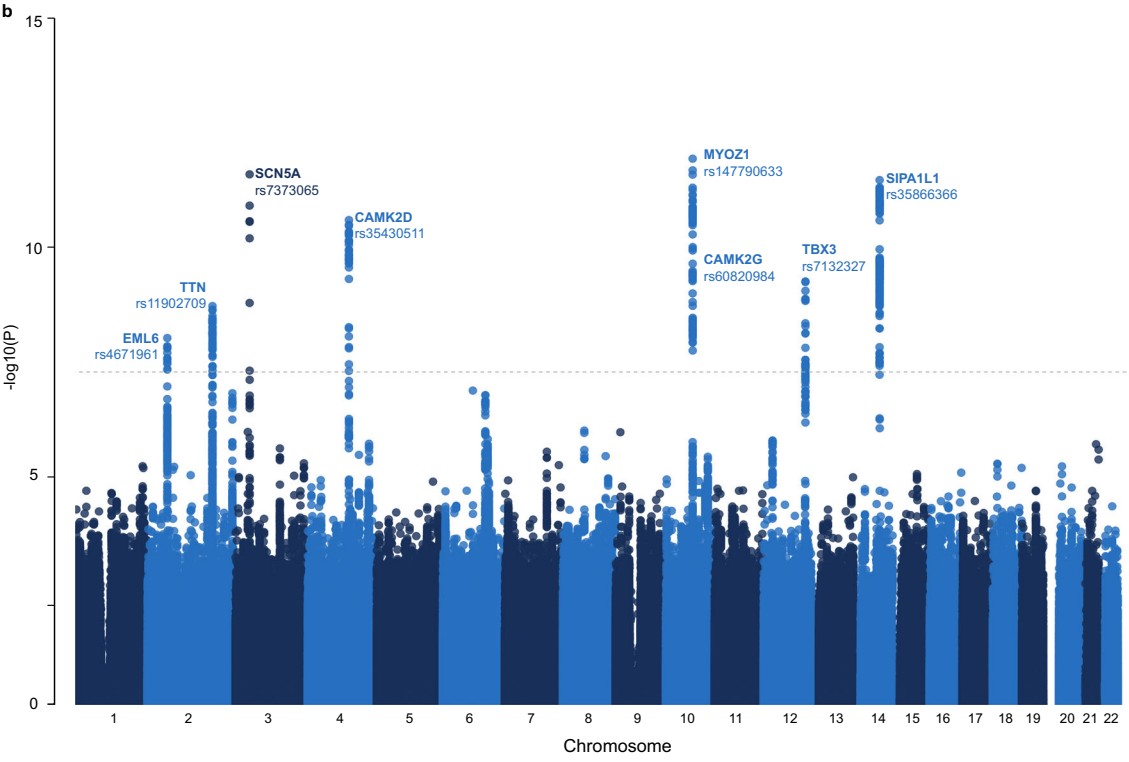

**Fig. 8 | Manhattan plots with the results of the genome-wide association study of image- and electrocardiogram (ECG)-derived age-deltas.** This figure shows the −log₁₀(P value) on the y axis across all autosomal chromosomal positions (x axis) derived from large-scale one-sided regression analyses using PLINK. The dotted line indicates genome-wide significance (P = 5 × 10⁻⁸), which accounts for multiple testing at genome-wide level. Significant loci are labelled by their most likely causal gene and the lead single nucleotide polymorphism (Table 1). **a** Imaging feature-derived cardiovascular age-delta GWAS (n = 29,506). **b** ECG-derived cardiovascular age-delta GWAS (n = 31,475).

**Table 1 | Summary of genetic association studies**

| Lead variant | | | | | GWAS result | | | | | Annotation | | | | Evidence |
| rsID | Chr | Ref | Alt | EAF | Estimate | SE | P | Locus genes | Closest gene | Most likely causal gene | Missense | eQTL | Mech | Evidence qual. |
|---|---|---|---|---|---|---|---|---|---|---|---|---|---|---|
| **Genome-wide association study results: image-derived cardiovascular age-delta** | | | | | | | | | | | | | | |
| rs2042995 | 2 | T | C | 0.24 | 0.43 | 0.08 | $1.3\times10^{-8}$ | TTN | TTN | TTN | Y | Y | Y | High |
| rs7795735 | 7 | T | A | 0.44 | 0.46 | 0.06 | $3.5\times10^{-13}$ | ELN | ELN | ELN | N | N | Y | Low |
| rs1991860 | 8 | C | T | 0.41 | −0.46 | 0.08 | $5.0\times10^{-9}$ | PI15 | PI15 | PI15 | N | Y | N | Medium |
| rs61886308 | 10 | G | A | 0.18 | 0.56 | 0.08 | $2.4\times10^{-11}$ | PLCE1 | LOC107984255, PLCE1 | PLCE1 | Y | Y | Y | High |
| rs2986036 | 10 | T | C | 0.47 | −0.36 | 0.06 | $9.5\times10^{-9}$ | NEURL1-AS1 | NEURL1-AS1 | NEURL1-AS1 | N | N | N | Low |
| **Genome-Wide Association Study Results: electrocardiogram-derived age-delta** | | | | | | | | | | | | | | |
| rs4671961 | 2 | G | A | 0.32 | 0.22 | 0.04 | $9.1\times10^{-9}$ | SPTBN1, EML6 | SPTBN1 | EML6 | N | Y | N | Medium |
| rs11902709 | 2 | C | T | 0.05 | 0.51 | 0.08 | $1.8\times10^{-9}$ | TTN, PLEKHA3, FKBP7 | TTN | TTN | N | Y | Y | High |
| rs7373065 | 3 | T | C | 0.02 | −0.93 | 0.14 | $1.2\times10^{-11}$ | SCN10A, SCN5A, EXOG | SCN5A | SCN5A | N | N | Y | High |
| rs35430511 | 4 | T | C | 0.25 | 0.27 | 0.04 | $2.5\times10^{-11}$ | CAMK2D | CAMK2D | CAMK2D | N | N | Y | High |
| rs147790633 | 10 | T | C | 0.14 | −0.37 | 0.05 | $1.2\times10^{-12}$ | AGAP5, BMS1P4-AGAP5, MYOZ1 | AGAP5 | MYOZ1 | N | Y | Y | High |
| rs60820984 | 10 | C | T | 0.19 | −0.28 | 0.05 | $1.8\times10^{-9}$ | CAMK2G, PLAU, NDST2 | CAMK2G | CAMK2G | N | Y | N | Low |
| rs7132327 | 12 | T | C | 0.27 | 0.25 | 0.04 | $5.4\times10^{-10}$ | TBX3 | TBX3 | TBX3 | N | N | N | Low |
| rs35866366 | 14 | A | G | 0.24 | 0.29 | 0.04 | $3.4\times10^{-12}$ | SNORD56B, SIPA1L1 | SNORD56B | SIPA1L1 | N | Y | N | Low |

**Rare variant association study results: image-derived cardiovascular age-delta**

| | Chr | EAF | Beta | SE | P | Gene |
|---|---|---|---|---|---|---|
| pLoF (AF <0.001/0.01) | 6 | $5.9\times10^{-4}$ | 5.74 | 1.25 | $4.4\times10^{-6}$ | TREM2 |
| pLoF (singletons only) | 8 | $9.5\times10^{-5}$ | −13.77 | 3.07 | $7.1\times10^{-6}$ | MICU3 |

Summary information on the lead single nucleotide polymorphisms (SNPs) of the genome-wide association study (GWAS)-identified significant loci and the genes identified from the rare variant association study (RVAS). For each significant locus, lead SNP summary information is provided (Chr chromosome, Ref reference allele, Alt alternative allele, EAF effect allele frequency of Alt), with GWAS summary statistics (Estimate, beta coefficient, SE standard error, P P value). Variant to gene annotation is provided, and a summary of the strength of evidence of gene mapping. Abbreviations: Missense (missense variant); eQTL status (co-localisation of GWAS signal with an expression quantitative trait loci for the gene in a plausible tissue type); Mech (the plausible mechanistic link between the gene and the phenotype, i.e. ageing); and Evidence qual. (Quality of evidence of variant to gene mapping, the strength of evidence deemed high, medium or low.). For RVAS, the EAF indicates the frequency of observing pLoF variants in the specified mask, the P values were obtained from one-sided burden tests on the additive effect, the significance level was set after adjusted by the number of genes in the tests ($P < 4.7\times10^{-6}$ for AF <0.001/0.01 and $P < 1.2\times10^{-5}$ for singletons). Analyses were performed independently for image-derived cardiovascular age-delta (n = 29,506) and electrocardiogram (ECG)-derived age-delta (n = 31,475). Gene and locus names are italicised.

related to chronological age or fully explained by later-in-life behaviours and exposures[59]. Age-delta provides information beyond variance in the individual traits used for prediction and is also independent of chronological age[60], with associations reflecting modifiers of healthy aging in the phenotypes assessed. We also observed that stronger effect sizes may be seen for age-delta SNP associations than underlying single traits. Longitudinal imaging will help to assess whether early-life influences and polygenic scores are the predominant drivers of the rate of aging across the lifespan and any pleiotropy between age-related changes in cardiac function and baseline traits.

We did not find that conventional ECG intervals improved predictive performance in a joint model with image-derived phenotypes, so we used an independent convolutional neural network to learn spatial features in ECG traces related to aging. Incorporating multimodal features that include imaging, ECG waveforms, and other clinical data could provide a broader assessment of traits contributing to cardiovascular aging, although this may be at the expense of model explainability and would need an appropriate network architecture. Our findings, using independent image and ECG models, also show distinct patterns and genetic associations of ageing in different systems suggesting that ageing is not a homogeneous process[9], and that conduction phenotypes provide complementary information to cardiovascular imaging on age-related atrial fibrosis and electrical remodelling. Further work is required to assess the contribution of processes to biological cardiovascular ageing not captured by CMR imaging or ECG data, for instance, atherosclerosis, by using more direct measures of plaque burden. Additionally, a longer follow-up period after image-phenotyping would provide greater power to investigate the association of age-delta with outcomes and the cumulative risks of accelerated aging, adjusting for any genetically associated disease phenotypes. Furthermore, understanding causal mechanisms and the effect of genetically-modulated cardiovascular ageing on 'health-span' could be performed on independent genotyped cohorts where time-to-event data is available. While we did not see a very strong association with binary outcomes, we may expect accelerated ageing to mediate age-associated morbidity as has been observed in other organ systems[46,61].

In conclusion, we found that cardiovascular ageing is linked to multiple modifiable risk factors, shows distinct patterns of remodelling, and is associated with loci related to genes regulating sarcomere homoeostasis, myocardial immunomodulation, and tissue responses to biophysical stress.

## Methods

All analyses in this study are available online (https://github.com/ImperialCollegeLondon/cardiovascular_ageing) and were conducted with R v.>3.6.0 and Python v.3.9.

### Participants

UK Biobank comprises ~500,000 community-dwelling participants aged 40–69 years who were recruited across the United Kingdom between 2006 and 2010[62]. All participants provided written informed consent for participation in the study, which was approved by the National Research Ethics Service (11/NW/0382). Our study was conducted under terms of access approval numbers 28807 and 40616. A range of available data were included in this study, comprising genotyping arrays and WES, cardiac imaging, health-related diagnoses, and biological samples. There were 488,252 genotyped participants, of which 454,787 have WES.

### Non-imaging phenotypes

Participants underwent a resting 12 lead ECG. Other phenotypes were collected by touch screen questionnaire, interview, biophysical measurement, hospital episode statistics, and primary care data (see Supplementary Material). Leucocyte telomere length (LTL) was measured using a multiplex qPCR assay from 488,415 available DNA samples of participants in UK Biobank[63]. After quality control (QC), valid LTL measurements were available for 472,577 individuals and log-transformed and z-standardised values were used for analysis (UK Biobank data field code 22192). Details of how each phenotype was acquired are available on the UK Biobank Showcase (http://biobank.ctsu.ox.ac.uk/crystal/).

### Imaging protocol

A standardised CMR protocol was followed to acquire two-dimensional, retrospectively-gated cine imaging on a 1.5T magnet (Siemens Healthineers, Erlangen, Germany)[64]. Short-axis plane cine imaging involved acquiring a contiguous stack of images from the left ventricular base to the apex, and long-axis cine imaging was also performed in the two and four-chamber views. Cine sequences all consisted of 50 cardiac phases with an acquired temporal resolution of 31 ms[64]. Transverse cine imaging of the ascending and descending thoracic aorta was also performed. Native T1 mapping within a single breath hold was performed at the mid-ventricular level using a shortened modified Look-Locker inversion recovery (ShMOLLI) sequence. Imaging phenotypes all underwent QC prior to use in analysis[65].

### Cardiac image analysis

Automated segmentation of the short-axis and long-axis cine images in UK Biobank was performed using fully convolutional networks[6]. Image segmentation in the UK Biobank dataset using this deep learning network is equivalent to expert human readers[65]. Volumes (end-diastolic, end-systolic, and stroke volume) and ejection fraction were determined for both ventricles. Myocardial volumes were used to compute left ventricular myocardial mass, assuming a density of 1.05 g ml$^{-1}$. Atrial volumes were calculated using the biplane area-length formula $V = \frac{8}{3\pi} \cdot \frac{A_{2Ch} \cdot A_{4Ch}}{L}$ (where $A_{2Ch}$ and $A_{4Ch}$ are the atrial areas on the two and four-chamber cine views, respectively, and $L$ is the averaged longitudinal diameter across two views). Measurements were indexed to body surface area (BSA) as per the Du Bois formula: $0.20247 \times (\text{Weight}^{0.425}) \times (\text{Height}^{0.725})$, with weight in kg and height in m. The heart was divided into 16 standardised anatomical segments, excluding the true apex[66].

The aorta was segmented using the cine images with a spatiotemporal neural network[67], from which maximum and minimum cross-sectional areas were derived. Distensibility was calculated using central blood pressure estimates obtained using peripheral pulse-wave analysis (Vicorder, Wuerzburg, Germany)[14].

Non-rigid image registration between successive frames enabled motion tracking on greyscale images.[68] Registration errors were minimised by tracking motion in both backwards and forwards directions from end-diastole, resulting in an averaged displacement field[14], which was then used to warp segmentations from end-diastole to successive adjacent frames. Circumferential (Ecc) and radial (Err) strains were calculated using short-axis cines as $E_{dir} = \frac{\Delta L_{dir}}{L_{dir}}$, where dir represents circumferential or radial direction, $L_{dir}$ the absolute length of a line segment along this direction and $\Delta L_{dir}$ its change in length over time. Longitudinal (Ell) strain was calculated from long-axis four-chamber motion tracking measured at basal, mid-ventricular, and apical levels. Segmental and global peak strains were then calculated. Strain rate was computed as the first derivative of strain, and thereafter peak early diastolic strain rate in radial (PDSR$_{rr}$) and longitudinal (PDSR$_{ll}$) planes was then identified. The ShMOLLI T1 maps were analysed using probabilistic hierarchical segmentation with automated quality control defining a region of interest within the interventricular septum as previously validated[16]. Blood pool T1 was used as a linear correction of myocardial T1 values[16,69].

Three-dimensional visualisation of left ventricular shape and motion with respect to age was performed through atlas-based

registration of the segmentations and motion fields. We calculated the mean phenotypes (shape, wall thickness and motion pathlines) for each decade of chronological age and represented these values on the epicardial surface of the models relative to participants aged 40–49 years[17].

## Cardiovascular age prediction

To define a model of healthy cardiovascular ageing, as previously validated in other organ systems,[11,61,70,71] we first partitioned the 39,559 participants into a development set, which consisted of 5063 "healthy" individuals that were free of cardiac, metabolic or respiratory disease, with a body-mass index below 30 (see Supplementary Material for details). We randomly split this group into separate training (80%, $n = 4019$) and test (20%, $n = 1044$) sets.

We used CatBoost, a decision-tree-based gradient boosting machine learning algorithm, on this feature set to predict the age of each participant in the healthy training and test sets. CatBoost was trained using the default hyperparameter set, except for early stopping rounds (patience) between 50 and 100, which was chosen by the default hyperparameter search method. A 10% validation holdout set ($n = 403$) was used for hyperparameter search (using the Python package Optuna). From the remaining training dataset, an additional 10% ($n = 362$) holdout set was taken to be used by CatBoost for internal early stopping evaluation (therefore, training CatBoost with the remaining $n = 3256$ instances). Thirty models (with distinct random initialisation seeds) were trained, and the one with the lowest mean absolute error on the holdout dataset was chosen.

The trained CatBoost model was used to predict the age of participants from the evaluation set ($n = 34,147$)[10,11]. As in brain age modelling[11,19,72–74], we corrected the correlation between age-delta and chronological age. For this bias-correction we used linear regression between the initial uncorrected cardiovascular age-delta against chronological age[73]. An offset is then calculated by multiplying chronological age by the slope of the regression line, and adding the intercept. A corrected predicted age is computed by subtracting this offset from the uncorrected predicted age.

## Evaluating associations of cardiovascular age-delta with risk factors

To explore associations of cardiovascular age-delta with classical cardiovascular risk factors, we curated four groups from the evaluation set, comprising individuals with obesity, hypertension, diabetes mellitus (DM) and coronary artery disease. We additionally obtained smoking frequency, serum levels of low-density lipoprotein, triglycerides and apolipoprotein B, z-adjusted leucocyte telomere length data, and derived alcohol consumption for participants (see Supplementary Material for details). For each of the four diseases, we selected a set of controls from the test set who did not have the disease, matched for age and sex using 1:1 propensity score matching. Each test sample thus had the same number of controls as the number with the disease of interest, and data from some participants also acted as control data for several disease groups (see Supplementary Material for further detail). Consistent with recommendations from brain ageing literature, all onward statistical analyses adjusted for age, age² and sex[10,74]. We then fit a linear regression model to test for the association between cardiovascular age-delta and the presence of disease. For continuous risk variables (alcohol and smoking consumption, serum levels of low-density lipoprotein, triglycerides and apolipoprotein B, and telomere length), we fit a linear regression model in all individuals of the test dataset that had complete data for these variables to assess the association between cardiovascular age-delta and risk factor. We adjusted for multiple comparisons by controlling the false discovery rate ($\alpha = 0.05$). For modelling associations with self-reported medication see Supplementary Materials.

## Outcome analysis

For assessing the association of cardiovascular age-delta with prospective cardiovascular events, we calculate the time-to-first major cardiovascular event (MACE) (stroke, heart failure, arrhythmia, cardiovascular death; see Supplementary Material for details) from the time of the imaging visit. We stratify by MACE prior to the MRI visit. Participants are split into quartiles, and associations with MACE are assessed through both descriptive analysis (cumulative incidence curves with all-cause death as a competing event) and model-based analysis with both a fully covariate-adjusted Cox model (age, sex, obesity, diabetes, smoking, alcohol consumption, hypertension, coronary artery disease and hypercholesterolaemia) or a minimally covariate-adjusted Cox model (age and sex).

## Medication effect analysis

We considered the most commonly used drugs, for which effectiveness on clinical outcomes is established: all major anti-hypertensives, i.e. beta-blockers, angiotensin-converting-enzyme inhibitors (ACEi), angiotensin receptor blockers (ARBs), calcium channel blockers (CCBs) and diuretics, as well as statins, metformin and digoxin (Supplementary Table 5). To assess the association between self-reported medication intake across the drugs and drug-classes considered, and cardiovascular age-delta, we performed separate drug-specific linear regression models with age-delta as the dependent variable, in 27,546 participants with complete data. We adjusted for sex, age, age², smoking pack years, alcohol consumption in grams per day, body-mass index, SBP/DBP, heart rate, prior diagnosis of obesity, coronary artery disease, hypertension, diabetes mellitus, hypercholesterolaemia and heart failure. All covariates were evaluated at the time of the imaging visit.

## Genotyping and sample quality control

Genotyping of UK Biobank participants has been detailed previously[62], and in brief, UK Biobank genotyping for 488,252 participants was performed on UK BiLEVE or UK Biobank Axiom arrays and imputation performed with the HaplotypeReference Consortium panel and the UK10K+1000 Genomes Project panel. We used UK Biobank Imputation V3 (in GRCh37 coordinates). The UK Biobank released whole exome data sequencing of 454,787 participants in 2021, and details regarding sequencing methods and variant calling procedures are described elsewhere[75]. We utilised genotypes in their released PLINK-format files, and restricted the cohort to the European population. We performed standard quality control steps recommended by UK Biobank, detailed in the Supplementary Material.

## GWAS analysis

GWAS analyses for cardiovascular age-delta were performed with PLINK (v.2) (n = 29,506). All GWAS analyses were adjusted for sex, age (at the time of MRI), age², the first ten genetic principal components, MRI assessment centre and genotyping array. Post-GWAS analysis removed SNPs with a Hardy–Weinberg equilibrium $P < 0.05$ and minor allele frequency (MAF) <0.005. Lead variants for each locus were assigned likely causal genes using variant annotations. Expression quantitative trait loci (eQTL) evidence for each locus was extensively searched for using the GTEx Portal and where available, full summary statistics were downloaded to assess co-localisation (Supplementary Material).

## Identification of rare variant gene-based associations that were independent of common variant signals

We performed rare-variants burden testing on UK Biobank WES using Regenie[26] on the Research Analysis Platform (RAP) (https://ukbiobank. dnanexus.com). The intersection of European participants with exome sequencing data, the age-delta phenotype, and array data after QC is 31,515. We further quality-checked the WES data by requiring that at

least 90% of genotypes for a given variant, independent of variant allele zygosity, have a read depth of at least 10 (the 90pct10dp filter) to avoid spurious hits. After QC, we ran step 1 to obtain predictors of individual trait values based on common genetic data, which were then used in step 2 for the rare variant gene burden testing.

For step 2, we annotated pLoF variants and defined gene variant sets using the UK Biobank 450K Exome helper files[76], also see UK Biobank Resource 916. We tested the pLoF variants on three separate burden masks per gene, based on the frequency of the alternative allele of the variants: MAF ≤1%, MAF ≤0.1% and singletons only. We tested genes with enough (5) pLoF carriers across all samples. Namely, the number of genes tested for singletons is 4271, for MAF ≤0.1% is 10,431, and for MAF ≤1% is 10,592. We performed Regenie step 2 with the adjustment of sex, age (at the time of MRI), age$^2$, the first ten genetic principal components, and genotyping array. The association was considered significant after multiple testing correction at $\alpha = 0.05$.

### Polygenic risk score (PRS) and PheWAS
Candidate variants for PRS for the cardiovascular age-delta were obtained based on the respective GWAS results by performing clumping (PLINK v1.9) using a linkage disequilibrium (LD) threshold of $R^2 = 0.1$ (in a window of 250 kb) and considering all SNPs with $P < 0.001$. These selected variants were then used to construct a genetic score for all individuals in the dataset using linear scoring in PLINK v2. Missing genotypes were imputed using the default mean imputation approach. We have additionally constructed a multivariable linear model evaluated on the European subset of the full imaging cohort, using sex, age (at the time of MRI), age$^2$, the first ten genetic principal components, MRI assessment centre and genotyping array as additional covariates, and cardiovascular age-delta as the dependent variable. We report the variance explained by the PRS as the difference of linear regression $R^2$ between a model of age-delta with all non-genetic covariates and the model that additionally includes the PRS as a covariate (see Supplementary Material).

PheWAS of image-derived cardiovascular age-delta and age-delta PRS was performed in European ancestry participants in the UK Biobank. PRS PheWAS was performed in participants that were not included in the derivation GWAS ($n = 373,948$), whilst image-derived age-delta PheWAS was performed in the same imaging cohort ($n = 34,137$). For phenotypes with at least 20 cases, the association was tested using logistic regression for categorical outcomes and linear regression for continuous traits, adjusting for age, age$^2$, sex, and the first ten genetic principal components. Statistical significance threshold ($P < 4.4 \times 10^{-5}$) was adjusted for multiple testing using Bonferroni correction for the total number of phenotypes tested (1149 phenotypes), and data presented with Manhattan plots grouped by body systems. PheWAS was performed using the `PheWAS` package in R version 4.0.3.

### Age prediction using resting electrocardiograms
Age-delta predictions have been performed previously using ECG data[77]. We were interested to see how predictions from CMR imaging and resting ECG data might correlate, and also investigate shared genetic variants for age-delta. To achieve this, we adapted a previously published neural network model to perform the ECG-based predictions. We trained the model using ECG input data from the UK Biobank RAP and re-formatted these according to the requirements of the model. Subsequently, we fine-tuned the pre-trained model using the same development set used in the imaging-based predictions. Details on data input adaption and model refinement are provided in Supplementary Material. We used the same statistical and GWAS approach for ECG age-delta as described for image-derived cardiovascular age-delta.

### Reporting summary
Further information on research design is available in the Nature Portfolio Reporting Summary linked to this article.

## Data availability
All raw and derived data in this study are available from UK Biobank (http://www.ukbiobank.ac.uk/), conducted under application number 40616. GWAS summary level data were publicly available through the GWAS catalogue (https://www.ebi.ac.uk/gwas/), deposited using accession numbers GCST90239748 and GCST90239749. For colocalization analyses, we used the unfiltered eQTL results from eQTL Catalogue (https://www.ebi.ac.uk/eqtl/) and the Genotype-Tissue Expression (GTEx) Portal v.8 (https://gtexportal.org/home/).

## Code availability
The code used for our analyses are publicly available: https://github.com/ImperialCollegeLondon/cardiovascular_ageing(https://doi.org/10.5281/zenodo.8143760)[78].

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

## Acknowledgements
D.P.O'R. acknowledges support from Bayer AG, the Medical Research Council (MC_UP_1605/13); National Institute for Health Research (NIHR) Imperial College Biomedical Research Centre; and the British Heart Foundation (RG/19/6/34387, RE/18/4/34215); J.S.W. acknowledges support from the Sir Jules Thorn Charitable Trust [21JTA]. This research has been conducted using the UK Biobank Resource under Application Numbers 28807 and 40616. The genetic association analyses were conducted on the UK Biobank Research Analysis Platform (https://ukbiobank.dnanexus.com). We thank James Cole (UCL) and Tobias Kaufmann (University of Tübingen) for advice on age-delta modelling. We also thank Tim Dawes, Ghalib Bello, and Carlo Biffi (Imperial College London) for developing the three-dimensional cardiac modelling used in this study. We also acknowledge Esther Puyól-Anton, Bram Ruijsink and Reza Razavi for the T1 mapping data.

For the purpose of open access, the authors have applied a creative commons attribution (CC BY) licence to any author-accepted manu-script version arising.

## Author contributions
M.S. and M.H.d.A.I. performed the formal analyses and co-wrote the paper; W.B. and A.P.K. performed the image analysis; K.A.M., E.E., I.K., C.L., J.S.W., J.M., C.B. and D.F.F. performed or interpreted the genetic and outcome analyses; S.L.Z., P.-R.S., A.d.M. and A.C. performed addi-tional data modelling; M.R.W. critically revised the manuscript; D.P.O'R. conceived the study, managed the project and revised the manuscript. All authors reviewed the final manuscript.

## Competing interests
J.M., E.E., I.K., C.B. and D.F.F. are full-time employees of Bayer AG, Germany. D.P.O'R. has received research support and consultancy fees from Bayer AG. The remaining authors declare no competing interests.

## Additional information

**Peer review information** *Nature Communications* thanks Dan Arking, Minglang Yin and the other, anonymous, reviewer(s) for their contribu-tion to the peer review of this work. A peer review file is available.

