## [Peer Review File · Nature Communications]

Environmental and genetic predictors of human cardiovascular ageingEditorial Note: This manuscript has been previously reviewed at another journal that is not operating a transparent peer review scheme. This document only contains reviewer comments and rebuttal letters for versions considered at *Nature Communications*.

REVIEWER COMMENTS

Reviewer #2 (Remarks to the Author):

I recommend this paper for publication.

Reviewer #3 (Remarks to the Author):

The authors have not really addressed the main questions that I raised in my previous review.

Ongoing/major concerns:

- 1) It is still unclear how much of the GWAS results are a function of the age-delta vs. the underlying traits that contribute to the age-delta. This can be directly tested by comparing results for age-delta to those for the underlying traits, with the expectation that the association will be stronger for age-delta than any of the underlying traits. One could also do mediation analysis, seeing if the association is driven by a particular contributing trait.
- 2) The ECG metric is still presented as an after-thought in the main text, with most of the relevant results in the supplemental material. This is quite surprising given that the correlation between this measure and chronological age is stronger than the cMRI measure.
- 3) The impact/interpretation of the difference between the cMRI age-delta and ECG age-delta is not adequately addressed. If you are really measuring cardiovascular aging by these methods, how is it they are completely uncorrelated (Supp. Fig 11, right)? I get that the underlying mechanisms leading to cardiovascular aging are likely differentially captured by these methods, but a complete lack of correlation implies each of these methods is measuring something different, so you cannot generalize and say that either of these methods is measuring cardiovascular aging. At best, each one is capturing a specific aspect of cardiovascular aging, and this argues very strongly that these measures likely need to be combined in some fashion to more appropriately measure cardiovascular aging.
- 4) The "Risk factors for cardiovascular aging" section is quite confusing. It seems like the PheWAS is $\text{trait} \sim \text{age.delta}$, whereas the risk factors are $\text{age.delta} \sim \text{risk.factor}$. These are fundamentally different hypotheses, and need to be presented as such. This section then lumps in the incident CVD events, and then moves on to drugs. This section needs to be broken up and expanded to clarify the hypotheses being tested and results. Additionally, the drug results are almost entirely in the supplement, and the main text does not give sufficient detail to interpret the results.
- 5) For the GWAS, there is no real explanation of how samples were selected to be "Discovery" or "Replication" samples. Also, I don't see a rationale for dividing up the sample, as power is greater using the full analyses, and indeed, it seems like the full analyses are the main results.

Reviewer #4 (Remarks to the Author):

In this study, the authors pursued a thorough analysis of the environmental and genetic factors that contribute to the discrepancies between chronological age and predicted age based on cardiovascular phenotypes. Data on cardiac structure and function, the cardiac conduction system, and the vascular system were derived from cross-sectional MRI and ECG data from the UK Biobank. These data were subjected to advanced machine learning techniques to create an "age-delta" phenotype, which was subsequently the focus of cross-sectional epidemiological analyses, common- and rare-variant association testing, and polygenic score analyses. The authors identified numerous clinical risk factors and rare/common genetic loci associated with the age-delta phenotype, demonstrating the

heterogeneity of cardiovascular aging, and highlighting potential roles for specific biological pathways and processes in normal versus accelerated cardiac aging.

This is a comprehensive and insightful analysis of the factors influencing human cardiovascular aging. The data sources are among the most robust available to date. The methodology employed is rigorous and all analyses have been conducted in a meticulous fashion. The authors have also done an excellent job of responding to the primary queries raised by Reviewers.

MINOR COMMENTS:

(1) While the noted strengths are many, there are continued questions surrounding the age-delta metric and its capacity to provide information beyond the individual measures used to generate this metric. As stated by others, several SNPs arising from the GWAS are for genes previously associated with aortic or ventricular phenotypes (i.e. ELN and TTN). To determine the added value of this derived value, perhaps the age-delta metric could be tested for association with future outcomes (i.e. MACE) adjusted for relevant aortic or ventricular phenotypes/pathologies (i.e. aortic dilation or cardiomyopathy)?

(2) The lack of imaging data on coronary (or other) atherosclerosis as inputs into the models producing the age-delta phenotype remains a concern. Despite the response to the Reviewers that atherosclerotic processes may be captured (to some extent) through aortic imaging, this almost certainly under-weights the contribution of atherosclerosis to CV aging as compared to the well-captured structural/functional and electrical factors. In addition to the text that has been added already, a more forthright acknowledgement of this in the Limitations section would be helpful.

(3) The cross-sectional associations between age-delta phenotype and various CV risk factors (like hypertension) are interesting. However, temporality and causality are difficult to determine/disentangle from cross-sectional assessments. Might, for example, Mendelian randomization be employed to determine whether genetic proxies for particular risk factors (namely, those with cross-sectional links) associate with the age-delta phenotype, thereby providing some suggestion of a causal/directional pathway?

Reply to reviewers' comments

We thank the reviewers for their very constructive comments on our manuscript. In the paper the first set of revisions is shown in red, and the current set of revisions in green.

Reviewer #2 (Remarks to the Author): I recommend this paper for publication.

Response: *We thank the reviewer for their previous suggestions to improve the paper and their recommendation.*

Reviewer #3 (Remarks to the Author): The authors have not really addressed the main questions that I raised in my previous review. Ongoing/major concerns:

1. It is still unclear how much of the GWAS results are a function of the age-delta vs. the underlying traits that contribute to the age-delta. This can be directly tested by comparing results for age-delta to those for the underlying traits, with the expectation that the association will be stronger for age-delta than any of the underlying traits. One could also do mediation analysis, seeing if the association is driven by a particular contributing trait.

Response: *We thank the reviewer for wanting more clarity in the paper on this. To recap, as shown across the broad brain-age literature, including comparable age-delta GWAS studies (Jonsson et al, Nat Commun, 2019; Vidal-Pineiro et al, eLife 2021), the aim is to index deviations from population norms across multiple parameters of "healthy" aging and find associations with sequence variants and risk factors. We can visualise these patterns of healthy aging across multiple imaging phenotypes relating to vascular function, interstitial fibrosis, cardiac remodelling, and diastolic function etc (Supplementary Fig. 2). The combination of these 126 IDPs into an estimate of biological age is a compact summary metric, and the route by which the most accurate estimate of a subject's age can be predicted from the imaging data available. Age is the greatest common risk factor for chronic diseases but age-related decline varies markedly between individuals - and this is what we aim to capture in using cardiovascular "age-delta" as a trait. Our discovered loci are related to genes regulating hallmarks of maladaptive aging that include fibrosis, cardiac remodelling, vascular degradation, and inflammation.*

To directly respond to the question of how much the results are a function of age-delta compared to the underlying traits, and to assess their relative strength of association, we performed a comparison between the age-delta GWASs and an underlying trait GWAS. We assessed diastolic function which plays a pivotal role in age-related disease, is causally associated with outcomes, and is a leading predictor of age-delta (Thanaj et al, Nat Cardiovasc Res, 2022). Taking an identical image-derived phenotype (radial peak diastolic strain rate) we found no overlap in SNPs significant at genome-wide level suggesting distinct mechanisms modify aging. Below we include a new figure showing that the standardised effect sizes for each of these discovered SNPs tend to be greater for both age-delta traits than an underlying trait. None of the trait SNPs had significant effect sizes in either age-delta GWAS. We also show no strong relationship between age-delta and the underlying trait.

As others have shown for equivalent real and simulated data (Smith et al, Neuroimage, 2019), and we show for our cardiac IDPs, age-delta provides information beyond variance in the individual traits used for prediction and is also independent of chronological age, so any discovered loci represent putative modifiers of healthy aging in the phenotypes assessed. Where loci are shared, or indeed not replicated, by trait GWASs reflects their effect on cardiovascular aging. Genetic analysis offers a way of exploring factors that influence phenotypic variation towards an improved understanding of lifespan trajectories in the cardiovascular system and we offer evidence that cardiac age-delta is a genetically influenced trait.

We have also provided an updated PheWAS using the PRS of cardiovascular age-delta in Supplementary Fig. 8. showing that genetic risk for cardiovascular aging is associated with a broad range of outcomes and biomarkers. The effect size plot shown below has been added as Supplementary Fig. 5. We have also added this to the Discussion:

“Age-delta provides information beyond variance in the individual traits used for prediction and is also independent of chronological age,⁶⁰ with associations reflecting modifiers of healthy aging in the phenotypes assessed. We also observed that stronger effect sizes may be seen for age-delta SNP associations than underlying single traits. Longitudinal imaging will help to assess whether early-life influences and polygenic scores are the predominant drivers of the rate of aging across lifespan and any pleiotropy between age-related changes in cardiac function and baseline traits.”

Supplementary Figure 5. Standardised effect sizes for significant SNPs. a. Hex plot of age-delta residuals and underlying trait z-scores. b. The absolute effect size per standard deviation for each genome-wide significant SNP across three independent GWASs. Each GWAS discovered different SNPs and are labelled by RSID. PDSR, peak diastolic strain rate; ECG, electrocardiogram; SD, standard deviation.

- The ECG metric is still presented as an after-thought in the main text, with most of the relevant results in the supplemental material. This is quite surprising given that the correlation between this measure and chronological age is stronger than the cMRI measure.

Response: Thank you for this suggestion. We have now incorporated ECG-derived age prediction into the main text. We now describe this in the “Introduction” section (“We also trained a model to learn spatial features of electrocardiograms (ECG) associated with aging”) and we have now provided the same performance metrics for both image-derived and ECG-derived age prediction in the main “Results” section (“There was a strong correlation between predicted age and chronological age ($|r| = 0.85$, $P < 2.2 \times 10^{-16}$) using latent ECG features which is comparable to other deep learning architectures. There was also no relationship to the participants’ actual age ($|r| = -0.01$, $P \approx 1$)). Additional phenotyping detail has also been added in the main “Results” section, and the paragraph header changed to “Image and electrocardiogram phenotyping”. The Supplement retains technical details of the model specification, data preparation and training. We have also expanded the discussion of the discovered loci (see Point 3 below).

“Latent features of electrocardiographic (ECG) traits may also be associated with aging and so we also trained a deep learning network to predict corresponding “ECG-age” and age-deltas to discover any shared genetic architecture with cardiovascular aging.”

“There was a strong correlation between predicted age and chronological age ($|r| = 0.85$, $P < 2.2 \times 10^{-16}$)

using latent ECG features which is comparable to other deep learning architectures.¹⁸ There was also no relationship to the participants' actual age ($|r| = -0.01, P \approx 1$).

The ECG-age GWAS and variant annotation results are already in the main section (Table 1 and Figure 4). There is also a section in the main methods describing "Age prediction using resting electrocardiograms".

3. The impact/interpretation of the difference between the cMRI age-delta and ECG age-delta is not adequately addressed. If you are really measuring cardiovascular aging by these methods, how is it they are completely uncorrelated (Supp. Fig 11, right)? I get that the underlying mechanisms leading to cardiovascular aging are likely differentially captured by these methods, but a complete lack of correlation implies each of these methods is measuring something different, so you cannot generalize and say that either of these methods is measuring cardiovascular aging. At best, each one is capturing a specific aspect of cardiovascular aging, and this argues very strongly that these measures likely need to be combined in some fashion to more appropriately measure cardiovascular aging.

Response: *Our reasoning for keeping the two models (image-based and ECG-based) separate relates to the two different approaches we applied for age prediction in each dataset and the value of understanding their shared and distinct aging mechanisms. We used well established image-derived phenotypes that are readily measurable, clinically interpretable and closely-coupled physiologically ie - cardiac volumes, vascular function, myocardial fibrosis etc. As shown in Figure 2 this allows us to show the features associated with aging and their relative importance in the CatBoost prediction model. We did train a joint model, using CatBoost, including image-phenotypes and conventional ECG intervals but this did not improve predictive performance (Supplementary Table 2). For that reason we trained a neural network to learn features associated with aging from the time-resolved ECG trace. The convolutional residual neural network architecture is not suited to a joint analysis with tabular image-derived data as there is no spatial relationship in the IDPs for the network to model. Using a neural network we would also not be able to meaningfully explain the predictions or assign feature importance.*

As the Reviewer highlights, ECG data captures features of aging that are complementary to the image-derived phenotypes. Recent work by others has highlighted how different organ systems age at different rates through distinct mechanisms (Tian et al, Nat Med, 2023), and our GWAS also suggests that, as might be expected, electrophysiological aging is in part mechanistically distinct to structural/functional aging in the wider cardiovascular system. For instance, as we have added to the Discussion, we found two genes (SCN5A and CAMK2D) associated with ECG age-delta that are implicated in atrial fibrosis and early onset AF. Common variants in TTN, which may mediate both atrial and ventricular fibrosis, are associated with both image-based and ECG age-deltas, and are recognised as having age-dependent effects that are distinct to the changes in isoform expression seen in heart failure. So while we have used a multiparametric assessment of cardiovascular aging, to gain power from modelling related traits and improve predictive performance, we would propose keeping the reporting of the ECG and image-derived models separate as this preserves model explainability, is technically tractable, and reveals shared and distinct genetic associations.

We have added these arguments to the Discussion, highlighting that there may be heterogeneous aging mechanisms in different physiological systems. We hope this clarifies our reasoning for the approach we took and provides critical discussion of the findings.

"Common variants in TTN were also implicated in ECG age-delta suggesting a shared genetic mechanism with conduction-associated traits, however other loci were predominately related to pathways regulating cardiac rhythm, suggesting that the conduction system ages through distinct mechanisms, as has been observed in other organs and tissues.⁴⁶ For instance, CAMK2D is a gene associated with atrial remodelling, fibrosis and premature atrial fibrillation, and we found it to be a factor influencing aging in the conduction system.⁴⁷ Fibrosis due to aging is thought to play a major role in modulating con-

duction and repolarization diseases, with age also known to affect the severity and expressivity of the channelopathy-associated gene *SCN5A* identified in the ECG age-delta GWAS.⁴⁸ Gain or loss of function mutations in *SCN5A* are also implicated in progressive atrial fibrosis although the mechanisms are not fully understood.⁴⁹”

“We did not find that conventional ECG intervals improved predictive performance in a joint model with image-derived phenotypes, and so we used an independent convolutional neural network to learn spatial features in ECG traces related to aging. Incorporating multi-modal features that include imaging, ECG waveforms and other clinical data could provide a broader assessment of traits contributing to cardiovascular aging, although this may be at the expense of model explainability and would need an appropriate network architecture. Our findings, using independent image and ECG models, also show distinct patterns and genetic associations of aging in different systems suggesting that aging is not a homogeneous process and that conduction phenotypes provide complementary information to cardiovascular imaging on age-related atrial fibrosis and electrical remodelling.⁹”

4. The “Risk factors for cardiovascular aging” section is quite confusing. It seems like the PheWAS is trait~age.delta, whereas the risk factors are age.delta~risk.factor. These are fundamentally different hypotheses, and need to be presented as such. This section then lumps in the incident CVD events, and then moves on to drugs. This section needs to be broken up and expanded to clarify the hypotheses being tested and results. Additionally, the drug results are almost entirely in the supplement, and the main text does not give sufficient detail to interpret the results.

Thank you for highlighting how this section could be improved. The “PheWAS” is an assessment of the association between age-delta (predictor) and a broad range of outcomes available in UKB, ie outcome~age.delta. We then aimed to investigate the associations between several established cardiovascular risk factors and age-delta. For this analysis, the regression was age.delta~risk.factor. We have now clarified the two approaches within a re-named section “Phenome wide association study and risk factor analysis” and have also expanded on the methodology in the Supplementary Materials (“PheWAS and risk factor analysis methods”). The Results text has also been clarified as follows:

“We first aimed to assess associations of cardiovascular age-delta with an unbiased range of diseases in the UKB using a PheWAS. Here we performed independent logistic regression analyses with age-delta as a predictor for each outcome adjusted for age, sex, ethnicity and multiple comparisons...”

“We next aimed to quantify the associations of established risk factors and cardiovascular age-delta. We used linear regression with each risk factor in turn as a predictor and age-delta as the dependent variable...”

We have further sub-divided the subsequent Results sections into “Age-delta and cardiovascular events” and “Prescribed medications and cardiovascular age”. The latter enables us to expand on the drug associations in the main text, including moving Fig. 4 to the main Results section, with further details of the linear regression modelling for this analysis in the Supplementary Materials.

We have also updated the PRS-PheWAS in Supplementary Fig. 8 by including a broader range of phenotypes, as well as providing full details of the model fitting.

5. For the GWAS, there is no real explanation of how samples were selected to be “Discovery” or “Replication” samples. Also, I don’t see a rationale for dividing up the sample, as power is greater using the full analyses, and indeed, it seems like the full analyses are the main results.

The data had been partitioned by an externally-defined date cut-off but we agree that as the main findings are from the full dataset, it makes sense not to divide the GWAS cohort and so we have updated Table 1 and the Methods accordingly.

Reviewer #4 (Remarks to the Author): In this study, the authors pursued a thorough analysis of the environmental and genetic factors that contribute to the discrepancies between chronological age and predicted age based on cardiovascular phenotypes. Data on cardiac structure and function, the cardiac conduction system, and the vascular system were derived from cross-sectional MRI and ECG data from the UK Biobank. These data were subjected to advanced machine learning techniques to create an “age-delta” phenotype, which was subsequently the focus of cross-sectional epidemiological analyses, common- and rare-variant association testing, and polygenic score analyses. The authors identified numerous clinical risk factors and rare/common genetic loci associated with the age-delta phenotype, demonstrating the heterogeneity of cardiovascular aging, and highlighting potential roles for specific biological pathways and processes in normal versus accelerated cardiac aging.

This is a comprehensive and insightful analysis of the factors influencing human cardiovascular aging. The data sources are among the most robust available to date. The methodology employed is rigorous and all analyses have been conducted in a meticulous fashion. The authors have also done an excellent job of responding to the primary queries raised by Reviewers.

MINOR COMMENTS:

1. While the noted strengths are many, there are continued questions surrounding the age-delta metric and its capacity to provide information beyond the individual measures used to generate this metric. As stated by others, several SNPs arising from the GWAS are for genes previously associated with aortic or ventricular phenotypes (i.e. ELN and TTN). To determine the added value of this derived value, perhaps the age-delta metric could be tested for association with future outcomes (i.e. MACE) adjusted for relevant aortic or ventricular phenotypes/pathologies (i.e. aortic dilation or cardiomyopathy)?

Response: *We thank the reviewer for acknowledging the strengths of our paper. In response to a comment from Reviewer #3 we have now performed additional comparisons of trait effect sizes showing that SNPs in the age-delta GWAS tend to have stronger associations than an underlying trait. As others have shown for equivalent real and simulated data (Smith et al, Neuroimage, 2019), and we show for our cardiac IDPs, age-delta provides information beyond variance in the individual traits used for prediction and is also independent of chronological age, so any discovered loci represent putative modifiers of healthy aging in the phenotypes assessed. Where loci are shared (ELN, TTN) by trait GWASs, or indeed not replicated reflects their effect on cardiovascular aging not variance in the traits. For example, TTN is recognised as having age-dependent effects that are distinct to the changes in isoform expression seen in heart failure and there is evidence that ELN expression and organization modifies arterial aging. The additional figure is shown below and the following text added to the Discussion:*

“Age-delta provides information beyond variance in the individual traits used for prediction and is also independent of chronological age,⁶⁰ with associations reflecting modifiers of healthy aging in the phenotypes assessed. We also observed that stronger effect sizes may be seen for age-delta SNP associations than underlying single traits. Longitudinal imaging will help to assess whether early-life influences and polygenic scores are the predominant drivers of the rate of aging across lifespan and any pleiotropy between age-related changes in cardiac function and baseline traits.”

We also thank the reviewer for their suggestion regarding outcome modelling. In the main paper we report associations between age delta and incident MACE (as defined in Supplementary Table 4), and found an association between with outcomes (hazard ratio 1.09, $p = 0.02$) after adjusting for major cardiovascular risk factors (including age, sex, obesity, diabetes, smoking, alcohol consumption, hypertension,

Supplementary Figure 5. Standardised effect sizes for significant SNPs. **a.** Hex plot of age-delta residuals and underlying trait z-scores. **b.** The absolute effect size per standard deviation for each genome-wide significant SNP across three independent GWASs. Each GWAS discovered different SNPs and are labelled by RSID. PDSR, peak diastolic strain rate; ECG, electrocardiogram; SD, standard deviation.

coronary artery disease, hypercholesterolaemia). Clinical diagnoses of cardiomyopathy are rare in UKB with a reported prevalence of 0.3%, making adjustment unreliable given the relatively small number of outcomes observed (860 events in our participants), which is why we used a range of established risk-factor covariates. This aspect could be addressed in future as the sample size grows and endpoints accumulate, which we now highlight in the limitations section of the Discussion.

“Additionally, a longer follow-up period after image-phenotyping would provide greater power to investigate the association of age-delta with outcomes and the cumulative risks of accelerated aging, adjusting for any genetically associated disease phenotypes.”

2. The lack of imaging data on coronary (or other) atherosclerosis as inputs into the models producing the age-delta phenotype remains a concern. Despite the response to the Reviewers that atherosclerotic processes may be captured (to some extent) through aortic imaging, this almost certainly under-weights the contribution of atherosclerosis to CV aging as compared to the well-captured structural/functional and electrical factors. In addition to the text that has been added already, a more forthright acknowledgement of this in the Limitations section would be helpful.

Response: *We agree with this suggestion and in addition to the existing text, have further added this point within the Limitations section as follows:*

“Further work is required to assess the contribution of processes to biological cardiovascular aging not captured by CMR imaging or ECG data, for instance atherosclerosis by using more direct measures of plaque burden.”

3. The cross-sectional associations between age-delta phenotype and various CV risk factors (like hypertension) are interesting. However, temporality and causality are difficult to determine/disentangle from cross-sectional assessments. Might, for example, Mendelian randomization be employed to determine whether genetic proxies for particular risk factors (namely, those with cross-sectional links) associate with

the age-delta phenotype, thereby providing some suggestion of a causal/directional pathway?

Response: *We agree that longitudinal data would help to understand temporal dependencies, and particularly whether risk factors result not only in higher, but accelerating, cardiovascular aging. As mentioned above, we found associations between age-delta and outcomes, but we felt that it was not strong enough to justify the assumptions for causal inference on the data currently available. Indeed, we might also expect genetic risk factors for accelerated aging to result in earlier onset of morbidity that may not be captured through conventional two-sample MR models or recorded in external case-control datasets. We have added the following to the Discussion:*

“...understanding causal mechanisms and the effect of genetically-modulated cardiovascular aging on “health-span” could be performed on independent genotyped cohorts where time-to-event data is available. While we did not see a very strong association with binary outcomes we may expect accelerated aging to mediate age-associated morbidity as has been observed in other organ systems.^{46,60}”

We have also expanded the PheWAS, using an age-delta polygenic risk score, to comprise a broad range of biomarkers, physiological measures and outcomes revealing 8 associated traits including hypertension and lung function (Supplementary Fig. 8).

Supplementary Figure 8. PheWAS using PRS of cardiovascular age-delta. Phenome wide analysis of age-delta PRS adjusted for age, age², sex and the first ten genetic principal components by logistic regression ($n = 415,622$ genotyped White British UK Biobank participants). The red line represents the significance threshold after accounting for multiple testing using Bonferroni correction (2407 traits, $P < 2.1 \times 10^{-5}$). Upright triangles indicate positive associations, and the inverted triangles indicate negative associations. FEV1, forced expiratory volume in first second.

REVIEWERS' COMMENTS

Reviewer #3 (Remarks to the Author):

The authors have appropriately addressed my previous comments.

Reviewer #4 (Remarks to the Author):

The authors have provided thorough responses to all of my queries and, in the process, further improved the manuscript. In particular, I appreciate the detailed demonstration of stronger associations between genetic variants with the age-delta phenotype than with certain underlying traits, which serves to bolster the utility of this approach. I have no further comments or requests.